# Differences in Maturity and Anthropometric and Morphological Characteristics among Young Male Basketball and Soccer Players and Non-Players

**DOI:** 10.3390/ijerph18083902

**Published:** 2021-04-08

**Authors:** Stefania Toselli, Francesco Campa, Pasqualino Maietta Latessa, Gianpiero Greco, Alberto Loi, Alessia Grigoletto, Luciana Zaccagni

**Affiliations:** 1Department of Biomedical and Neuromotor Sciences, University of Bologna, 40126 Bologna, Italy; stefania.toselli@unibo.it (S.T.); alessia.grigoletto2@unibo.it (A.G.); 2Department of Life Quality Studies, University of Bologna, 47921 Rimini, Italy; francesco.campa3@unibo.it (F.C.); pasqualino.maietta@unibo.it (P.M.L.); 3Department of Basic Medical Sciences, Neurosciences and Sense Organs, University of Study of Bari, 70121 Bari, Italy; 4School of Pharmacy, Biotechnology and Sport Science, University of Bologna, 40126 Bologna, Italy; alberto.loi10@studio.unibo.it; 5Department of Neuroscience and Rehabilitation, Faculty of Medicine, Pharmacy and Prevention, University of Ferrara, 44121 Ferrara, Italy; luciana.zaccagni@unife.it; 6Center of Sport and Exercise Sciences, University of Ferrara, 44123 Ferrara, Italy

**Keywords:** sport practice, adolescent, physical characteristics, exercise science, biological maturation

## Abstract

*Background*: An aspect that influences sport performance is maturation status, since, within the same chronological age group, boys who have advanced maturation outperform their late maturing peers in tests of muscular strength, power, and endurance. Therefore, the aims of the present study were: (i) to investigate the differences in biological maturation and anthropometric and morphological characteristics among three groups of Italian adolescents, two of which were sportive (practicing basketball and football) and one non-sportive, and (ii) to identify the anthropometric and morphological predictors that best discriminate these three groups. *Methods*: Sixty-one basketball and 62 soccer players and 68 non-sportive youths were measured (mean age = 13.0 ± 1.1 y). Anthropometric characteristics were taken and body mass index, cormic index, body composition parameters, and somatotype were derived. An estimation of maturity status was carried out considering the years from peak height velocity (PHV). Two-way 3 × 3 ANOVAs was performed on all anthropometric characteristics to test the differences within sport groups and maturity status groups. Discriminant function analysis (stepwise criteria) was then applied to anthropometric and body composition variables to classify subjects into the three different sport categories. *Results*: Differences in anthropometric characteristics were detected among the three groups. For somatotype, differences among all of the considered groups were higher for endomorphy (*p* < 0.001; effect size = 0.13). Biological maturity influences the differences in the anthropometric characteristics and body composition among subjects of the same chronological age during adolescence. The variables that best discriminated the three groups were represented by body composition parameters, body proportions, and body build. *Conclusions*: This study confirms that boys who practice sport present healthier body composition parameters, with lower level of fat parameters. The assessment of maturity status is a fundamental factor in explaining anthropometric and body composition differences among peers in this period. Its comprehension may assist coaches and technical staff in optimizing competitive efficiency and monitoring the success of training regimes.

## 1. Introduction

Soccer and basketball are among the most played team sports worldwide [1,2]. Both are team sports that depend on a combination of high levels of physical, technical, and tactical skills. Their popularity is due to players’ athleticism, expressed by an optimal combination of body size, physique, motor abilities, and technical skills [1,3,4]. These features are often represented as multivariate profiles associated with success in response to training and competition, not only in senior, but also in youth players [5]. The anthropometric characteristics are decisive for an optimal physical level and, therefore, a good level in the game, and can be different depending on the type of sport practiced and on the game position [6,7,8]. Measurements of individuals’ linear dimensions, body diameters, body mass, and skinfolds, which are considered the key components of the somatic build profile of an elite player, have been monitored [6,9]. These measurements are critical in selecting players for each position on the court [10,11,12]. Arnason et al. [13] have shown that the anthropometrical profile of the soccer players has been associated with measures of match-related performance, and reported that teams with the highest fitness levels and the lowest percentage of body fat (%BF) had a higher league ranking. A high percentage of body fat is negatively associated with velocity over 20 m, an important determinant variable in the performance of soccer players [14,15]. The appropriate amount of %BF can be reasonably considered to be around 10% [9]. Soccer players have higher percentages of muscle in comparison to the sedentary population, being as high as 62% [12,16]. The importance of a basketball player’s body size is well-documented: anthropometric characteristics, such as body fat, skinfold thickness, body height, arm span, and body circumferences, were determined to be the principal components in elite basketball players; therefore, they are often regarded as indicators of the level of play [1,17,18,19].

In addition to anthropometric characteristics, morphology constitutes an important and reliable indicator of players’ performance, and somatotype represents the elective method to evaluate it. Somatotype is defined as the quantification of the present shape and composition of the human body, and it is accepted as one of the indicators of physical body structure [20,21,22]. The ideal somatotype for an athlete differs according to the requirements of the sport and the different playing positions [2,21]. Various studies have shown that the most predominant somatotype component of soccer players is balanced mesomorph with high muscularity and a low fat percentage [7,23]. This predominance in mesomorphy has also been reported in young soccer players, although their ectomorphy is higher than in adult players [2]. Male basketball players are likely to display a mesomorph somatotype: professional players from top teams can present mixed (meso-ectomorphic) and balanced somatotypes [19].

Considering youth teams, it is well-documented that, within the same chronological age group, boys who have advanced maturation outperform their late maturing peers in tests of muscular strength, power, and endurance [24,25,26]. On average, more mature boys and girls are taller and heavier than their peers of the same chronological age, but with a later maturation; this gives them a huge advantage in sports that include physical contact. It is between 11 and 16 years of age that we find the greatest variation in maturational status between youngsters. There is a strong link between maturational development and growth and performance [27]. Organizing the groups of players on the basis of chronological age, without considering maturity status, leads to big differences in size, body composition, and performance among athletes. Biological maturation has a large impact on selection procedures. Regarding this aspect, Malina et al. [28] described the biological variability in the developmental period and its consequences on body size and shape: until 13 years of age, the average maturity is the most represented maturity category, and those who mature late and early tend to be equally represented, while in the following age groups there are almost no delayed players. With age and a probable increase in experience, players with a more advanced maturity tend to dominate the game; data suggest that sport systematically excludes more delayed players in favor of those who are average and, especially, of those who are advanced [27].

Given that some of the available results are inconsistent, there still exists a strong call to better nurture youth soccer and basketball players, especially during the pubertal period, when maturity status can impact a player’s growth, development, and performance. According to Gryko et al. [19], the somatotype and other anthropometric variables might be specific to geographical region, especially during growth and maturation, and studies regarding Italian sport youth are limited. Considering the current state of knowledge in this field, it might be beneficial to examine a large extent of anthropometric data, such as breadths and circumferences, which are related aspects of body build. Moreover, the importance of this period for youth development is unquestionable, since systematic differences in maturity status among individuals of the same chronological age are well-known [29], and there is a need to have information regarding the percentage of subjects in the different maturity categories in sportive and non-sportive groups and on their characteristics. In addition, the influence of maturity status on anthropometric, somatic, and body composition characteristics has been observed in sport youth, but not in non-player subjects.

Thus, the aims of the present study were (i) to investigate the differences in biological maturation and anthropometric and morphological characteristics of the sportive youngster (practicing basketball and soccer) versus non-sportive youths, and (ii) to identify the minimum set of predictors that best discriminate these groups, so that this can be used by coaches and technical staff to improve the development and selection of young athletes, as well as increase the opportunities in their training session and competition. 

We expect that adolescents who practice sport present peculiar characteristics that distinguish them both between each other and with non-players, especially in body composition parameters, and that maturation status influences physical dimensions and body composition. 

## 2. Material and Methods

### 2.1. Participants and Design

A cross-sectional study was carried out in a sample of 191 pre-adolescent boys (age: 13.01 ± 1.15 years) living in the city of Bologna (Emilia Romagna region, North Italy): a group of 61 adolescents attending basketball teams (in the age categories under 12, under 13, under 14, and under 15) of the youth academy of the professional Italian basketball club—Fortitudo Bologna—registered in the maximum Italian division, a group of 64 children participating in soccer teams (from the under 12 to under 15 age categories) of the youth academy of the professional Italian soccer club Bologna Football Club 1909 registered in the maximum Italian championship and a control group of 68 middle-school students who did not practice any sports. All of the soccer and basketball players belonging to under 12 through under 15 teams were considered. The mean years of sport practice were comparable between the two sports (6.7 ± 2.2 in basketball and 6.9 ± 2.4 in soccer). The non-players were recruited from a centrally located school in the city of Bologna.

Regarding basketball, in addition to tournament matches, the players trained for 4.5 h a week (three workouts of 1.5 h each), while soccer players trained for 6 h a week (four workouts of 1.5 h each).

All of the subjects volunteered to participate in the study, giving their verbal assent. Written informed consent was provided by the parents before the study began. The study was approved by the Bioethics Committee of the University of Bologna (approval code: 25027).

### 2.2. Procedures

Anthropometric characteristics (height, weight, lengths, widths, circumferences, and skinfold thicknesses) were collected by a trained operator according to standardized procedures [30]. Height and sitting height were measured to the nearest 0.1 cm using a stadiometer (GPM, Zurich, Switzerland), and leg length was derived by the subtraction of sitting height from height. Body weight was measured to the nearest 0.1 kg (light indoor clothing, without shoes) using a calibrated electronic scale. Circumferences (relaxed and contracted upper arm, waist, thigh, and calf) were measured to the nearest 0.1 cm with a non-stretchable tape and widths (humerus and femur) to the nearest 0.1 cm with a sliding calliper. Skinfold thicknesses (biceps, triceps, subscapular, supraspinale, sovrailiac, thigh, and calf) were measured to the nearest 1 mm using a Lange skinfold calliper (Beta Technology Inc., Houston, TX, USA). Each anthropometric characteristic was carried out three times, and the mean value was considered.

Body mass index (BMI) was computed as weight (kg)/stature squared (m^2^), and the cormic index was calculated as sitting height (in centimetres) × 100 divided by height (in centimeters), which means sitting height as a proportion of stature.

Body composition parameters (percentage of fat mass (%F), fat mass (FM, kg), and fat-free mass (FFM, kg)) were calculated using the skinfold equations that consider the combination of triceps and subscapular skinfolds developed by Slaughter and colleagues [31]. Fat mass index (FMI, kg/m^2^) was derived as FM (kg)/stature squared (m^2^), and fat-free mass index (FFMI, kg/m^2^) was computed as FFM (kg)/stature squared (m^2^). The total area (cm^2^) of the upper arm (TUA), calf (TCA), and thigh (TTA), the muscle area (cm^2^) of the upper arm (UMA), calf (CMA), and thigh (TMA), and the fat area (cm^2^) of the upper arm (UFA), calf (CFA), and thigh (TFA) were calculated according to Frisancho [32]. In addition, arm fat index (AFI), calf fat index (FCI), and thigh fat index (TFI) were derived.

Somatotype components (endomorphy, mesomorphy, ectomorphy) were calculated according to the Heath−Carter anthropometric method [20].

### 2.3. Maturity Status

An estimation of the years from peak height velocity (PHV), which is an indicator for the adolescent growth spurt, was made using the equation for boys developed by Mirwald and colleagues [33].
Maturity offset = −9.236 + 0.0002708 (leg length × sitting height) − 0.001663 (age × leg length) + 0.007216 (age × sitting height) + 0.02292 (weight/height).

Since maturity offset represents the time before or after PHV, the years from PHV were calculated by subtracting the age at PHV from chronological age.

Malina & Koziel [34] reported that the approximation of the age at PHV (APHV), based on the prediction equation used, is often lower in younger children who are not yet in their adolescent growth spurt, and higher in older and sexually mature participants who already passed their adolescent growth spurt. To overcome this potential age effect, we followed the approach proposed by Rommers et al. [35], who used age-specific *z*-scores to classify players according to their maturity status. The predicted APHV was used to calculate *z*-scores within each specific age category (U10−U15, *n* = 6). Based on these age-specific *z*-scores of the predicted APHV, players were then classified as “earlier” (*z* < −1), “on-time” (−1 ≤ *z* ≤ 1), or “later” (*z* > 1) in maturing [36].

### 2.4. Statistical Analysis

Variables normality was verified with the Kolmogorov−Smirnov test. Descriptive statistics (means and SD) were calculated. Percentage frequency was determined for qualitative variables (maturity status).

Differences in the frequencies were tested by the chi-squared test, with Fisher’s exact test and Bonferroni correction pair-wise comparison. Two-way 3 × 3 ANOVAs were performed on all anthropometric characteristics to test the differences between sport groups and maturity status groups. When a significant F ratio was obtained, the Tukey post hoc test was used to evaluate the differences among groups. Effect sizes using partial eta squared (η^2^) were calculated and interpreted using the benchmarks provided by Cohen (0.01 = small, 0.06 = medium, and 0.14 = large) [37]. Discriminant function analysis (stepwise criteria) was then applied to anthropometric and body composition variables to classify subjects into the different sport categories. 

The data analysis was performed using Statistica for Windows version 8.0 (Stat Soft Italia SRL, Vigonza, Padua, Italy).

## 3. Results

The division of the sample according to maturity status for each category of sport showed that over two-thirds of the boys were on time in maturing. Soccer players had the lowest rate of late maturing subjects (9.68%) and basketball players the highest of late maturers (16.39%); the non-players group had the highest rate of early maturing (19.12%), but the differences were not significant (χ^2^ = 1.36; *p* = 0.850).

Table 1 shows the anthropometric characteristics of the sample according to sports practice (soccer, basketball, and no sport) and maturity status (early, on-time, and late). Height, weight, sitting height, and leg length presented significant differences both within sports and maturity status groups, and medium to large effect sizes were identified. All of the mean values increased with the progression of maturity status. Regarding differences within sports groups, on average, basketball players always showed the highest values for all of the considered parameters. Soccer players generally presented higher mean values than the non-sportive groups, with significant differences particularly between early mature soccer players and on-time or late non-players.

Cormic index demonstrated the difference in body proportions among the boys of different maturational status, increasing with maturation. A large effect size was detected. Significant differences were observed between late maturing soccer players (who presented the mean lowest values) and early or on-time mature players of all of the sports groups.

In general, circumferences increased with maturity status and presented significant differences within maturity groups, except thigh circumference, which showed significant differences also within sports groups. Moderate effect sizes were identified within sport groups, while there were large effect sizes considering maturity status. Early maturing basketball players had the highest mean values for all of the circumferences (except the thigh one, for which the highest was in early mature soccer) and the late mature non-sportive boys had the lowest mean values.

Mean values of humerus and femur widths significantly differed within maturity status (with a large effect size), and femur width also differed among sports groups, presenting a large effect size. A greater diameter was observed in early maturing basketball players, which significantly differed from all of the late maturing subjects and with soccer and non-sportive on-time maturing participants.

Regarding skinfolds thicknesses, the greatest differences were observed within sports groups: soccer players presented the lowest mean values and non-sportive participants the highest ones. The effect sizes confirmed the differences within sports groups. Early and on-time non-sportive participants presented significant differences with soccer players. Consequently, percentage fat and fat mass index were significantly the lowest, and fat free mass index was the highest in soccer players. The areas of the limbs followed the same trend, as testified by AFI, CFI, and TFI (Table 2).

Regarding somatotype, significant differences within sports groups were noticed for endomorphy, for which a large effect size was detected. As for the other fat parameters, soccer players presented the lowest values, with significant differences with non-sportive groups: early maturing soccer players versus non-sportive on-time maturing participants; late maturing soccer players versus early and on-time non-sportive participants and, finally, on-time soccer players versus early and on-time non-sportive participants and versus on-time basketball players.

Mesomorphy and ectomorphy, on the contrary, showed significant variations within maturation groups.

Figure 1 shows the somatoplots of the considered groups, taking into account maturity status. The influence of maturation on somatotype is evident in all of the groups. In particular, late maturing participants of all of the groups presented a higher ectomorphy than the other groups of maturation, being ectomesomorphs. Early and on-time basketball players and non-sportive subjects were mesomorphs, while soccer players of the same categories presented a higher ectomorphic component.

Stepwise discriminant analysis identified nine predictor variables (Table 3). Wilks’ lambda denotes how useful a given variable is in the stepwise analysis and determines the order in which the variables enter into the analysis. TFI entered into the discriminant analysis first, followed by CFI, height, cormic index, FFMI, humerus width, %F, endomorphy, and, lastly, femur width. The discriminant function was significant (Wilks’ lambda = 0.2623, *p* < 0.0001), indicating that the selected variables differentiated well between subjects of the three groups. By this function, 62.3% of basketball players, 83.8% of non-sportive subjects, and 85.5% of soccer players were correctly classified.

Table 4 shows the standardized coefficients for canonical variables. The high discriminant power of body composition parameters (%F, endomorphy, and FFMI) in the first canonical variable and CFI, height, femur width in the second one is notable; the first accounted for 83.1% of the variance, and the second 16.9%. Two discriminant functions were obtained: the first function presented a lower Wilks’ lambda (value = 0.262), and the test was highly significant (*p* < 0.001). The plot of the two discriminant functions confirmed the separation of the three groups (Figure 2).

Figure 2 represents the canonical analysis and group centroid distances among sports categories for the two discriminant functions.

## 4. Discussion

The first aim of the present study was to investigate the differences in biological maturation and anthropometric and morphological characteristics of the sportive youngster (practicing basketball or soccer) versus non-sportive youth.

Considering differences connected to sports practice, basketball players of the present study presented the highest dimensions regarding height, weight, circumferences, and widths, while soccer players showed the lowest values relative to fat parameters. Non-sportive groups generally presented the highest values for fat parameters and the lowest values for FFMI. This confirms our hypothesis on the difference in body characteristics and composition parameters, both between sportive youth and with non-sportive. The relevance of anthropometric characteristics in young athletes is thus already evident during adolescence, with specific features for each sport. These differences are important, since anthropometry affects performance. The importance of basketball players’ body sizes, specifically tall stature, is well-documented; in fact, basketball players are typically taller than the players of other games [1,38]. Tests on both young and professional players revealed that individuals who were taller in stature and had longer limbs obtained higher scores regarding efficiency on the court and achieved better physiological parameters [19]. Basketball requires handling the ball above the head; therefore, having a greater height is an advantage in this sport. Regarding soccer players, our results are in accordance with the literature: generally, soccer players are mainly distinguished for body composition parameters, having higher percentages of muscle and lower fat percentage in comparison to the sedentary population [9]. According to Boraczyński et al. [39], the fluctuations in body mass are generally rare in young male soccer players, who are characterized by a slender body build in comparison to their non-training counterparts. Moreno et al. [40] found significantly lower BF% in male soccer players aged 9.0–14.9 years in comparison with non-training boys.

Regarding somatotype, differences among groups were observed only for endomorphy: it confirms what was observed for fat parameters, being that this component is lower in soccer players and higher in non-sportive youths. Somatotype values of Italian basketball players participating in our study were prevalent in mesomorphy. Generally, the somatotype of basketball players was specified as balanced mesomorph or ectomorphic mesomorphy [3,41]. Thus, the findings of our study agree with other studies regarding the mesomorph component. The mesomorph component also obtained higher scores regarding efficiency on the court [19]. In soccer players, mesomorphy is usually the most predominant somatotype component. Generally, the somatotype of elite soccer players is balanced mesomorph, and the predominance in mesomorphy has also been reported in young soccer players [42], although their ectomorphy is higher than in adult players [7,12,28]. 

In our hypothesis, maturation status influences physical dimensions and body composition both in sportive and non-sportive youth. While the influence of maturation on physical characteristics has been observed in sportive youth, no data were available for non-sportive. The results of the present study showed that many differences are explained by the maturity status, since early maturing participants of all of the groups presented a dimensional superiority in comparison with the other two groups. Therefore, the maturity status is confirmed to be an element characterizing the differences within groups also in non-players, where the percentages of the three maturity statuses were similar to those of players. Height, weight, circumferences, and widths were greater in early maturing participants. In addition, they presented higher values of FFMI. In this case, the superiority of basketball players emerges for all of the maturity categories. Soccer players of all of the maturity groups presented lower values of parameters connected to adiposity. The body composition of soccer players is strongly influenced by maturity status: an increase in soft tissue and fluid is primarily related to somatic maturation [43]. Regarding body proportion, late maturing participants of all of the groups presented lower values of cormic index, confirming the differential timings of growth in height, sitting height, and leg length that occur with maturation. The growth of leg length precedes the PHV, while sitting height growth occurs after PHV [24].

This is also confirmed by somatotype, since an ectomorph mesomorph or ectomorphic mesomorph somatotype was observed for all of the late maturing participants. This is in accordance with what was reported by Malina et al. [24] on the relationship between morphology and maturation: late maturing is generally characterized by higher linearity than subjects with a more advanced maturation. Early and late mature soccer players presented a higher ectomorphic component than basketball and non-sportive groups. This is probably due to the lower skinfold thicknesses and, as consequence, endomorphy. The slender body build observed in the present study should be regarded as a positive manifestation of the soccer players’ somatic development, since other studies showed that the mesomorphic type of body build was dominant in young male soccer players with more frequent ectomorphy than in adult players [39]. As already reported, the balanced mesomorph somatotype has been observed for most positions in national and international studies categories on soccer players, but Gil et al. [44] found a decrease in the ectomorphy component with advancing age. Nikolaidis et al. [45] showed that the child group (U-12) differed from all of the age groups between U-13 and U-21 concerning endomorphy. According to the authors, somatotype components changed across adolescence: endomorphy and ectomorphy decreased, while mesomorphy increased. The results of the present study partially agree with the last statements, since it finds confirmation only for ectomorphy.

In the light of the above, this study confirms how biological maturity is a determining factor in influencing the differences in the anthropometric characteristics and body composition among subjects of the same chronological age during adolescence, both in subjects practicing sport and in non-sportive subjects. The importance of this period for youth development is unquestionable, as systematic differences in maturity status among individuals of the same chronological age occur [29]. In sports, by adolescence, team selection becomes less inclusive, and relies mostly on selection at team try-outs. Questions have arisen over the lack of due consideration to the potential impact of key growth and maturation processes occurring during adolescence on the selection process [46,47], and the consequences on sports participation for those deselected from sports teams. The bias found in youth sports, particularly male sports, favoring taller and stronger athletes means that the individuals with greater biological ages are more likely to be selected on sports teams [48]. The greater size and strength of earlier versus later maturing adolescents may mask, or be mistaken for, greater sport-specific skill [49]. In fact, along with the concerns of many researchers [50,51,52], the International Basketball Federation—World Association of Basketball Coaches (FIBA-WABC) [53] highlights that coaches working with 13–14-year-old youngsters must expect that, at this age, some players appear to be physically bigger while passing through a stage of great emotional vulnerability. Therefore, coaches must understand that some players improve faster than others, and must try to adapt to this [53]. In youth basketballers, tall stature is often the first marker for selection [52,54], and, as detected also in the present study, early maturing presents an advantage. Taken together, these results suggest that coaches should be careful when selecting players only based on anthropometric attributes, as they may simply be related to an advanced maturity status [52,55,56]. Regarding soccer, several professional soccer clubs and league-governing bodies (e.g., the English Premier League) have invested in the development of research-informed, long-term athlete development frameworks that account for the influence of biological maturity [57]. In younger soccer groups, the selection is more favorable for athletes who have matured early, while this factor may no longer be important as they approach adulthood. Furthermore, the boys selected due to their advanced maturation in the minor categories might then be rejected during the subsequent selection processes [43].

The second aim was to identify the minimum set of predictors that best discriminate the sportive and non-sportive groups in order to provide important and useful information that may help coaches to improve the development and selection of young players, as well as increase success opportunities in their training sessions and competitions. The variables that best discriminated the three groups were, in order of importance, TFI, CFI, height, cormic index, FFMI, humerus width, %F, endomorphy, and femur width. This means that body composition parameters, body proportions, and skeletal widths were the variables that maximized the differences between these three groups. Basketball players were distinguished by being the tallest and having larger skeletal diameters, while non-sportive youth showed the highest values of adiposity, a characteristic that differentiates them above all from soccer players. Gryko et al. [6] reported the importance of taking into consideration also the circumference and width measurements in determining the somatic profile of basketball players, since they provide additional information about a strong body build and adequate body musculature. Rebai et al. [58] reported 12–18% higher bone mineral density values in female basketball players than those observed in sedentary controls, and suggested that this sport produces high ground reaction forces estimated at about 3.53 to 5.74 times body weight when landing from a jump and a peak resultant braking force at about 1.35 times body weight when performing a run and rapid stop. In fact, high bone strain rates and magnitude are produced during basketball training and competition. This solicitation is considered to be effective strain stimulus to induce an osteogenic response. According to their opinion, this was a direct consequence of basketball participation rather than of self-selection. On the other hand, the body composition of a soccer player has a high impact on his performance and several studies have shown high correlation between the body fat percentage and athletic performance, %F being a determinant factor in elite players, who show a very low level [59].

Knowledge of body size and body composition profiles together with maturity status is an important step towards optimizing the selection and reducing de-selection of young players and promoting long-term development of player performance.

The cross-sectional design of the present study does not allow inferring about cause and effect, so longitudinal studies on Italian adolescent youth are needed to confirm and better understand the results of the present study. Future studies may consider the anthropometric and body composition of players in a longitudinal design, including maturation and functional tests, to make comparisons based on anthropometric characteristics.

## 5. Conclusions

Boys who play sports have lower adiposity than non-players. In basketball players, being taller and having a strong build are resulted distinctive characteristics, while in soccer players, this is represented by having a low %F. The morphological peculiarities that characterize a specific sport are clearly defined in this period.

The assessment of maturity status is a fundamental factor in explaining anthropometric and body composition differences among peers in this period. This aspect influences both sportive and non-sportive youth characteristics. In sports, its comprehension may assist coach and technical staff in optimizing competitive efficiency and monitoring the success of training regimes. We identified predictors that best discriminate the sportive and non-sportive groups, in order to provide important and useful information that may help coaches to improve the development and selection of young players, as well as to increase success opportunities in their training sessions and competitions.

## Figures and Tables

**Figure 1 ijerph-18-03902-f001:**
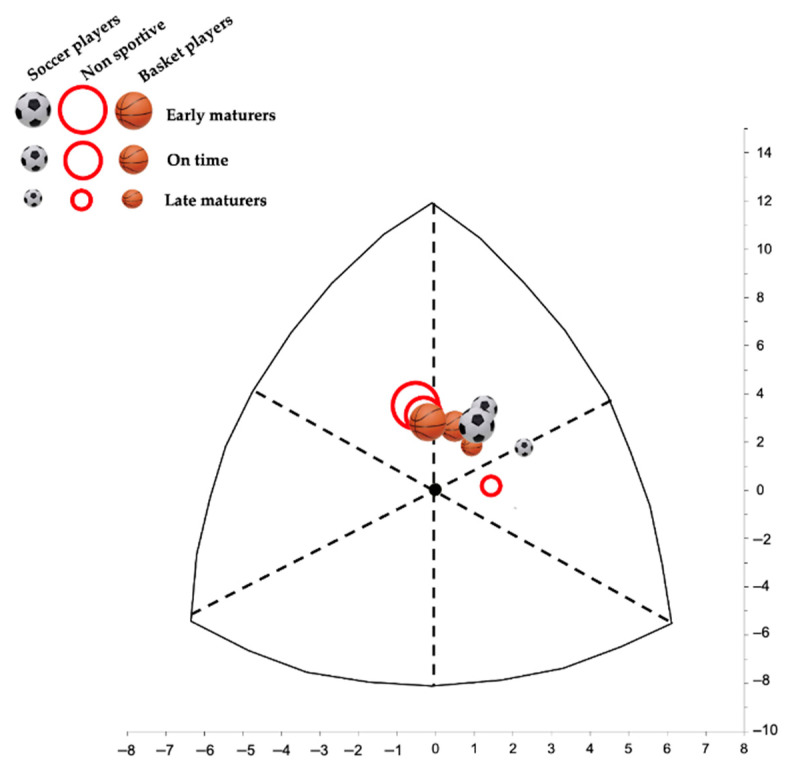
Somatoplots of the sportive and non-sportive groups according to maturity status.

**Figure 2 ijerph-18-03902-f002:**
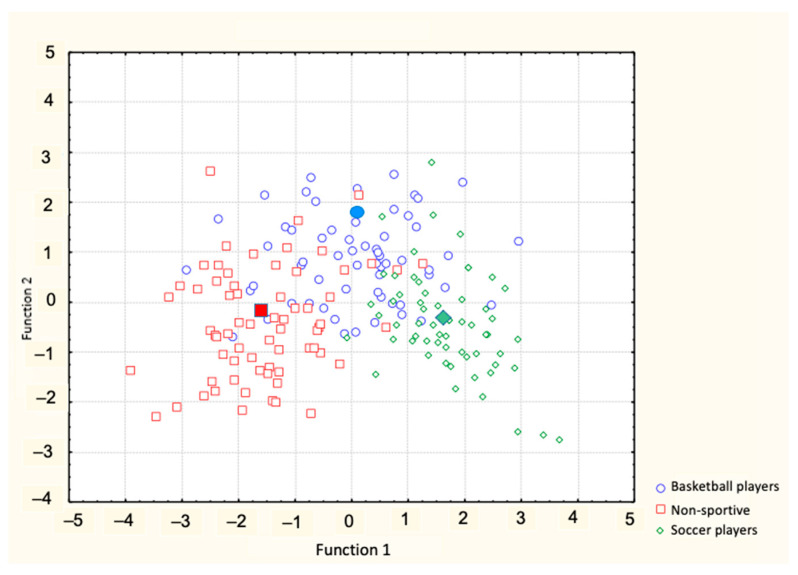
Canonical variate analysis: scatterplot of the canonical functions (full markers represent the centroids).

**Table 1 ijerph-18-03902-t001:** Anthropometric characteristics by maturity status and sports groups: results of ANOVA analysis.

	Basketball Players						Non-Players						Soccer Players						ANOVA					
	Early Maturers (*n* = 10)		Average Maturers (*n* = 41)		Late Maturers (*n* = 10)		Early Maturers (*n* = 13)		Average Maturers (*n* = 46)		Late Maturers (*n* = 9)		Early Maturers (*n* = 11)		Average Maturers (*n* = 45)		Late Maturers (*n* = 6)		Sports		Maturity		Sport * Maturity	
Traits	Mean	SD	Mean	SD	Mean	SD	Mean	SD	Mean	SD	Mean	SD	Mean	SD	Mean	SD	Mean	SD	F	*p*. η^2^	F	*p*. η^2^	F	*p*. η^2^
Height (cm)	173.6	11.5	166.1	9.7	155.7	13.4	167.1	5.7	154.8	7.9	148.4	8.5	171.5	11.5	159.0	13.5	155.1	14.8	6.5 **	0.07	21.1 ***	0.19	0.5	0.01
Sitting height (cm)	89.1	6.0	84.5	5.3	77.7	6.5	86.2	2.6	79.5	3.6	74.9	3.5	87.5	4.3	80.2	6.4	74.5	8.1	5.5 **	0.06	37.5 ***	0.29	0.4	0.01
Leg length (cm)	84.6	6.7	81.7	5.2	77.9	8.3	80.9	3.7	75.3	4.8	73.5	5.2	84.1	9.2	78.8	8.6	80.6	7.2	6.9 **	0.07	7.6 **	0.08	0.7	0.02
Weight (kg)	67.7	17.7	56.9	9.8	46.1	15.0	58.2	6.8	47.4	10.9	36.6	7.8	63.3	9.5	50.6	10.5	43.8	12.9	7.9 ***	0.08	26.8 ***	0.23	0.2	0.00
Cormic index	51.3	1.6	50.8	1.2	49.9	2.1	51.6	0.9	51.4	1.2	50.5	0.8	51.1	2.9	50.5	2.4	47.9	1.3	5.1 **	0.05	8.7 ***	0.09	1.2	0.02
BMI (kg/m^2^)	22.1	3.4	20.5	2.4	18.5	2.9	20.8	1.7	19.6	3.6	16.5	2.6	21.4	1.4	19.6	1.6	18.1	2.1	2.9	0.03	14.6 ***	0.14	0.4	0.01
Rel. arm circ. (cm)	26.2	2.1	24.8	2.9	22.3	4.1	25.2	2.1	23.8	3.2	20.9	2.7	25.6	2.1	22.9	2.5	21.4	3.4	2.1	0.02	14.7 ***	0.14	0.5	0.01
Contr. arm circ. (cm)	27.6	2.1	26.2	2.9	24.0	4.2	27.1	1.9	25.5	3.4	22.3	2.8	27.6	2.2	25.0	3.0	23.3	3.3	1.1	0.01	13.4 ***	0.13	0.4	0.01
Waist circ. (cm)	74.2	8.1	70.8	5.7	66.8	8.9	72.8	4.9	68.3	7.8	61.4	6.3	73.4	4.2	67.1	4.6	65.5	7.2	2.5	0.03	14.4 ***	0.14	0.7	0.02
Thigh circ. (cm)	50.0	6.9	46.5	3.9	43.5	6.2	48.2	2.8	44.7	5.3	39.1	3.9	50.1	2.8	45.3	4.4	42.3	6.7	3.4 *	0.04	19.6 ***	0.18	0.5	0.01
Calf circ. (cm)	36.7	3.9	34.7	2.8	31.9	4.0	35.6	1.9	33.3	3.3	29.5	2.1	35.6	3.9	33.6	4.0	31.2	3.4	2.4	0.03	15.7 ***	0.15	0.2	0.00
Humerus width (cm)	6.7	0.6	6.6	0.6	6.1	0.8	6.6	0.4	6.3	0.5	5.8	0.5	6.5	0.3	6.2	0.4	5.9	0.5	1.8	0.02	12.7 ***	0.12	0.6	0.01
Femur width (cm)	9.9	0.6	9.6	0.6	9.1	0.9	9.7	0.4	9.2	0.6	8.5	0.4	9.6	0.5	9.1	0.5	8.5	0.3	8.6 ***	0.09	24.0 ***	0.21	0.3	0.01
Biceps sk. (mm)	7.0	2.1	8.6	3.8	7.9	3.8	6.4	3.4	6.4	2.7	6.3	3.5	4.9	1.9	4.4	1.7	2.8	0.5	16.1 ***	0.15	0.9	0.01	0.9	0.02
Triceps sk. (mm)	9.3	2.9	10.6	3.5	10.4	3.2	11.1	4.8	11.0	3.5	10.4	4.2	8.3	2.6	8.1	2.4	7.3	1.7	7.7 ***	0.08	0.4	0.00	0.3	0.01
Subscap. sk. (mm)	8.7	2.3	8.7	2.9	6.9	2.4	9.0	3.9	9.2	5.4	6.6	2.1	7.6	1.7	5.9	1.3	5.2	0.5	3.9 **	0.04	3.3 **	0.03	0.5	0.01
Suprasp. sk. (mm)	11.9	5.5	9.3	3.4	7.4	3.0	12.3	5.2	11.1	6.6	8.4	8.3	6.5	2.3	5.2	1.6	3.4	0.9	14.7 ***	0.14	4.9 **	0.05	0.1	0.00
Suprailiac sk. (mm)	13.1	3.9	12.8	4.2	10.6	3.3	13.1	7.1	12.6	7.3	9.3	6.3	9.3	2.9	8.0	2.7	6.6	1.4	7.1 **	0.07	2.7	0.03	2.7	0.00
Medial calf sk. (mm)	10.4	2.8	10.9	3.5	10.1	3.2	12.2	3.5	10.6	3.6	9.8	3.5	7.6	3.2	6.9	1.9	5.0	1.5	21.1 ***	0.19	2.2	0.02	0.7	0.02
Lateral calf sk. (mm)	11.7	2.5	10.9	2.9	10.9	2.2	13.5	4.1	12.0	3.8	10.6	2.2	8.3	2.6	7.5	1.8	7.0	1.3	23.0 ***	0.20	2.6	0.03	0.4	0.01
Thigh sk. (mm)	13.4	3.6	13.3	4.1	13.0	3.5	17.3	5.6	16.1	5.6	14.7	4.7	10.9	3.6	10.2	2.5	9.3	2.2	18.1 ***	0.17	0.9	0.01	0.2	0.00
Sum of sk. (mm)	85.4	19.0	85.1	23.9	77.1	18.2	94.9	32.0	88.9	32.8	76.2	28.4	72.9	17.9	65.2	13.2	55.0	5.0	17.4 ***	0.16	2.9	0.03	0.2	0.00

Rel. arm circ. = relaxed arm circumference; contr. arm circ. = contracted arm circumference; circ. = circumference; sk. = skinfold; subscap. = subscapular; suprasp. = supraspinal. * *p* < 0.05; ** *p* < 0.01; *** *p* < 0.001, *p*. η^2^ = partial η^2^.

**Table 2 ijerph-18-03902-t002:** Body composition parameters and somatotype by maturity status and sports groups: results of ANOVA analysis.

	Basketball Players						Non-Players						Soccer Players						ANOVA					
	Early Maturers (*n* = 10)		Average Maturers (*n* = 41)		Late Maturers (*n* = 10)		Early Maturers (*n* = 13)		Average Maturers (*n* = 46)		Late Maturers (*n* = 9)		Early Maturers (*n* = 11)		Average Maturers (*n* = 45)		Late Maturers (*n* = 6)		Sport		Maturity		Sport * Maturity	
Traits	Mean	SD	Mean	SD	Mean	SD	Mean	SD	Mean	SD	Mean	SD	Mean	SD	Mean	SD	Mean	SD	F	*p*. η^2^	F	*p*. η^2^	F	*p*. η^2^
%F	15.7	3.0	16.2	4.8	15.1	3.7	15.9	5.0	16.4	4.7	15.2	4.8	13.4	4.7	12.7	3.7	11.5	2.2	5.4 **	0.06	1.4	0.02	0.2	0.00
FM (kg)	10.7	3.8	9.3	2.8	7.1	3.3	9.3	4.0	7.6	3.9	5.0	2.8	8.3	2.7	6.2	1.6	4.9	1.1	7.0 **	0.07	10.8 ***	0.11	0.2	0.00
FMI (kg/m^2^)	3.5	0.9	3.4	1.1	2.8	0.9	3.3	1.3	3.1	1.5	2.3	1.2	2.9	1.1	2.5	0.8	2.0	0.3	4.1 *	0.04	4.3 *	0.05	0.2	0.00
FFM (kg)	57.0	14.4	47.7	8.3	39.0	12.1	49.0	3.7	39.8	7.7	31.6	5.7	55.0	9.7	44.4	10.5	38.9	12.2	7.9 ***	0.08	25.4 ***	0.22	0.1	0.00
FFMI (kg/m^2)^	18.6	2.7	17.2	1.7	15.7	2.3	17.5	0.5	16.5	2.3	14.2	1.6	18.7	3.0	17.4	2.6	15.8	2.3	3.4 *	0.04	12.9 ***	0.12	0.1	0.00
TUA (cm^2^)	54.9	9.1	49.6	11.5	40.9	15.3	50.8	8.4	46.0	12.8	35.4	9.6	52.6	8.3	42.2	9.1	37.3	11.6	2.1	0.02	12.9 ***	0.12	0.5	0.01
UMA (cm^2^)	43.5	8.8	37.2	8.7	30.2	13.1	37.6	4.7	33.5	8.6	25.2	6.9	42.5	8.2	33.6	9.3	30.1	11.6	3.2 *	0.03	14.6 ***	0.14	0.5	0.01
UFA (cm^2^)	11.4	3.3	12.4	4.8	10.7	3.8	13.2	6.3	12.5	5.3	10.2	4.8	10.0	3.1	8.6	2.3	7.2	1.3	6.2 **	0.06	2.1	0.02	0.4	0.01
AFI (%)	21.0	6.9	24.7	6.7	27.3	9.1	25.0	8.8	26.5	6.2	28.4	9.0	19.4	6.1	21.2	6.9	20.8	7.1	7.2 **	0.07	2.2	0.02	0.3	0.01
TCA (cm^2^)	108.4	23.3	96.5	15.6	82.1	21.4	101.2	11.3	89.3	17.9	69.5	9.9	101.7	22.0	91.2	24.2	78.1	16.9	2.3	0.02	13.9 ***	0.13	0.1	0.00
CMA (cm^2^)	90.3	21.7	78.6	14.9	67.1	21.8	80.5	7.1	72.4	14.3	55.8	9.0	88.8	22.1	80.3	24.3	70.6	16.6	3.8 *	0.04	10.2 ***	0.10	0.2	0.00
CFA (cm^2^)	18.1	5.0	18.0	6.0	15.0	4.2	20.7	6.5	16.9	6.4	13.7	5.2	12.9	5.1	11.0	2.7	7.5	2.0	18.5 ***	0.17	6.8 **	0.07	0.6	0.01
CFI (%)	17.0	4.8	18.8	5.7	19.3	6.7	20.1	4.7	18.8	5.5	19.7	6.5	13.2	5.9	12.7	4.2	10.0	3.6	21.0 ***	0.19	0.1	0.00	0.9	0.02
TTA (cm^2^)	202.2	56.5	173.1	30.1	153.7	44.2	186.2	21.5	160.9	39.3	123.1	25.4	200.1	22.5	164.8	31.9	145.1	46.8	3.4 *	0.04	18.3 ***	0.17	0.4	0.01
TMA (cm^2^)	170.0	50.6	143.8	28.9	126.9	41.6	146.5	14.4	126.4	29.3	95.9	20.7	173.9	25.0	142.7	31.0	126.1	42.9	8.3 ***	0.08	17.0 ***	0.16	0.3	0.01
TFA (cm^2^)	32.2	10.0	29.3	9.1	26.8	7.3	39.7	13.8	34.5	15.0	27.2	9.9	26.2	7.9	22.1	5.3	19.0	5.6	11.0 ***	0.11	4.6 *	0.05	0.3	0.01
TFI (%)	16.2	4.6	17.2	5.2	18.1	5.4	21.0	5.8	21.1	5.5	22.1	6.4	13.3	4.7	13.7	3.6	13.4	3.4	23.8 ***	0.21	0.3	0.00	0.1	0.00
Endomorphy	3.0	0.9	3.0	1.0	2.7	0.7	3.3	1.3	3.4	1.5	2.9	1.4	2.2	0.8	2.0	0.6	1.6	0.3	13.1 ***	0.13	1.4	0.02	0.1	0.00
Mesomorphy	4.2	1.4	4.3	1.3	4.0	0.9	4.6	0.6	5.0	1.4	3.7	1.0	4.0	1.3	4.3	1.2	3.6	1.0	1.2	0.01	3.6 *	0.04	0.6	0.01
Ectomorphy	2.9	1.3	3.2	1.3	3.6	1.0	3.0	0.9	3.0	1.8	4.3	1.5	3.3	1.1	3.2	1.2	3.9	0.9	0.4	0.00	4.0 *	0.04	0.5	0.01

%F = percentage of fat mass; FM = fat mass; FFM = fat-free mass; FMI = fat mass index; FFMI = fat-free mass index; TUA = total upper arm area; UMA = upper arm muscle area; UFA = upper arm fat area; AFI = arm fat index; TCA = total calf area; CMA = calf muscle area; CFA = calf fat area; CFI = calf fat index; TTA = total thigh area; TMA = thigh muscle area; TFA = thigh fat area; TFI = thigh fat index. * *p* < 0.05; ** *p* < 0.01; *** *p* < 0.001, *p*. η^2^ = partial η^2^.

**Table 3 ijerph-18-03902-t003:** Selected variables by discriminant analysis.

Step	Entered	Wilks’ Lambda	Partial Lambda	F-Remove	*p*	Tolerance
1	TFI	0.284	0.925	7.251	0.001	0.458
2	CFI	0.284	0.924	7.337	0.001	0.474
3	Height	0.275	0.953	4.420	0.013	0.399
4	Cormic index	0.302	0.868	13.572	<0.001	0.547
5	FFMI	0.370	0.709	36.678	<0.001	0.312
6	Humerus width	0.282	0.931	6.587	0.002	0.347
7	%F	0.401	0.654	47.328	<0.001	0.120
8	Endomorphy	0.394	0.665	45.094	<0.001	0.115
9	Femur width	0.276	0.952	4.525	0.012	0.395

TFI = thigh fat index; CFI = calf fat index; FFMI = fat-free mass index; %F = fat percentage.

**Table 4 ijerph-18-03902-t004:** Standardized coefficients for canonical variables and chi-square test with successive roots removed.

Variable	Root 1	Root 2
TFI	−0.437	−0.392
CFI	−0.251	0.668
Height	−0.197	0.590
Cormic index	−0.562	0.378
FFMI	1.187	−0.325
Humerus width	−0.532	−0.243
%F	2.002	1.064
Endomorphy	−2.060	−0.819
Femur width	−0.284	0.512
Eigenvalue	1.794	0.365
Cumulative proportion	0.831	1.000
Canonical correlation	0.801	0.517
Wilk’s lamda	0.262	0.733
Chi-square	244.905	56.885
Degree of freedom	18	8
*p*	<0.001	<0.001

TFI = thigh fat index; CFI = calf fat index; FFMI = fat-free mass index; %F = fat percentage.

## Data Availability

The data that support the findings of this study are available from the corresponding author, upon reasonable request.

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
