# Peer review of "Differences in Maturity and Anthropometric and Morphological Characteristics among Young Male Basketball and Soccer Players and Non-Players"

_ijerph, 2021, doi:10.3390/ijerph18083902_

Round 1
Reviewer 1 Report
Overview
The authors recruited three groups of young adolescents that were specified as basketball players (n=61), soccer players (n=64), or non-sport participants (n=68). The groups were evenly matched for early, average, and late maturation. All subjects were measured for a battery of anthropometric assessments, body composition, indices (BMI, Cormic, index), and somatotype for comparisons between sport category. The findings confirm that biological maturity is associated with anthropometrical and body composition differences for youths within the same age groups. The authors were able to discriminate the categories using the variables fat index for the calf and thigh, height, the Cormic index, fat-free mass index, %fat, endomorphy, and widths for the humerus and femur. The authors conclude that sport participation has an effect on achieve optimal body composition and structure in adolescence, and that the indices they’ve measured may help coaches assess effectiveness of training programs and selection process for participants in soccer or basketball.
Major Concerns
The authors state the value of sport to achieve an optimal body composition, i.e., cause and effect. However, they have not considered the natural selection process – youths who are lean and more biologically mature than peers are more likely to succeed in physical activities and join and continue participation on sports teams. Height especially for a sport like basketball is a key discriminator; no amount of practice or training will stimulate a taller height in a player, it’s all in the natural selection of who inherently will grow taller and succeed in the sport. The participation part may have little to do with the biological maturation or body composition. More details in the Specific Suggestions section.
More transparency is needed on who the subjects are, from where they were recruited (same city or region of Italy), their socioeconomic status. The issue here, as always with a cross-sectional study, is whether sport alone can be suggested as the differentiator of the body composition, structure, or physical maturation of the subjects.
Specific Suggestions
Abstract
- lines 28-29 “For somatotype, differences among groups were higher for endomorphy:” This is confusing. Do the authors mean athletic groups?
- Lines 32-33, “Conclusion. This study confirms the value of sport to achieve an optimal body composition in adolescence.” I don’t believe the authors have the data to support this, nor does the study design – a cross-sectional observation – allow the conclusion to be made. Also “optimal body composition” for what? Health? Fitness, A specific sport?
Methods
- Line 111 “a group of 64 children attending soccer:” Attending soccer what? Team practice, camps, games?
- How were these camps, teams, and school selected? Does the recruitment process guarantee that the groups are simple random samples?
- Lines 115-117: For how many years had the youths in the sports groups been training? Same for the sports, on average?
- Lines 118-120: How did the minors volunteer? Written informed consent? Or verbal assent?
- Line 123-138: Was each anthropometric assessment – each of the lengths, height, skinfold sites and circumferences - measured only once?
- Line 138: As this reviewer recalls, the paper by Slaughter et al reported several skinfold combinations for predicting body fat. Which was used for this study and which of the seven skinfold sites measured in this study were used for the calculation?
Results
Page 8 of 16, line 1: As this reviewer understands it, the Cormic Index has a more specific definition than presented in the footnote; that it is the ratio of the sitting height to the total height2 (sitting height as a proportion of stature).
Discussion
Page 12 of 16 lines 75-77, “Non-sportive groups generally presented the highest values for fat parameters and the lowest values for FFMI, highlighting the value of sport to achieve an optimal body composition, also in young athletes [36]:” The authors are making a cause-and-effect conclusion, which is not justified with a cross-sectional design such as used in this study. Children typically gravitate to activities in which they have success or they may be invited/recruited to participate in activities for which the coach or parent sees a relevant body shape or natural ability in the children. A longitudinal study that follows the subjects from the same starting point to the same future point is required for this conclusion.
Page 13 of 16, lines 124-126 “…this study confirms how biological maturity is a determing (sic) factor in influencing the differences in the anthropometric characteristics and body composition among subjects of the same chronological age during adolescence.”
- Check spelling of determining.
- Since biological maturity was estimated using the same characteristics to express the physical stature, somatotype, and body composition, isn’t this connection a given (bias)? If a standard method of assessing biological maturation was used (testosterone levels of Tanner staging), I think this would be a fair statement.
Page 14 of 16, Line 163 in reference to basketball players “…and the most robust…:” What does this mean? The reviewer encourages the authors to stick with quantifiable outcomes, e.g., fat-free mass or whatever is meant by robust. Skeletal robustness, used a sentence earlier, should be defined in the methods.
Conclusion
Lines 166-167: “Sports practice is confirmed to be a key element to ensure a correct composition in adolescence.” This study doesn’t support this. See concerns listed above.
Reviewer 2 Report
TITLE:
Although the title is not inappropriate, but it could be more adjusted. Suggestion: “Differences in maturity, anthropometric and morphological characteristics among young basketball and soccer players and non-players”.
I think it is essential to talk about gender.
ABSTRACT
It can be improved. A part of the information that appears in the background, should be in the methods (participants and groups). The statistical analysis must be included, succinctly and objectively.
The expression “This study confirms the value of sport to achieve an optimal body composition in adolescence” it is a little abusive. Replace the word "optimal" for example with "better" or “most appropriate”.
Critical analysis of the Introduction
I think that the thematic framework of the main study variables should improve. sSome phrases are not properly cited. Ex: “Their popularity is due to players´ athleticism, expressed by an optimal combination of body size, physique, motor abilities, and technical skills”, and “The anthropometric characteristics are decisive for an optimal physical level and, therefore, a good level in the game, and can be different depending on the type of sport practiced and on the game position”, and more.
It should highlight the gaps that exist and emphasize the relevance of this study. You should also proceed with at least one hypothesis of the study, when formulating the objective (s) of the study. The definition of the study objective (s), should guide the entire study.
Critical analysis of the method
The chapter 2. “Materials and Methods” must have 4 sections (Participants; Instruments; Procedures; Statistical analysis). In my opinión it is important introduce the procedures.
Statistical procedures: For samples with a size greater than or equal to 30, it is advisable to test Kolmogorov-Smirnov. For samples of a smaller size it is the Shapiro-Wilk test is most suitable.
This can condition the inferences that come from this analysis. Also check the magnitude-effect values (for example, according to Wiersma & Jurss, 2003): 0.90 to 1.00 "Very high"; 0.70 to 0.90 "High"; 0.50 to 0 , 70 "Moderate"; 0.30 to 0.50 "Low"; 0.10 to 0.30 "Small".
Critical Analysis of Results
Table 1 seems to me to be an accessory. It has many tables which makes the article very "heavy".
Tables should have more important information (remove accessory information), with more content (ex: p value, Cohen's d). Take into account the comments of the statistical analysis, if you choose to accept the indications, the results will have to be based on those options.
If possible, join tables 4, 5 and 6. Try to simplify and make the reading more objective.
Critical Analysis of the Discussion
It could be more critical. Review this chapter again.
It would be interesting if they included possible justifications based on studies / authors on the (possible) reason for the results found. It would be good to have the 3 indicators (studies with similar results, studies with contradictory results and possible justifications). The suggestions for future studies should appear in the discussion. It must also have practical implications (Extremely important).
Critical analysis of the Conclusions
The conclusion is recommended to interpret the results in the light of the available evidence.
In this sense, this section could be improved. Do not make statements or inquiries that the study cannot support.
In the phrase “Sports practice is confirmed to be a key element to ensure a correct composition in adolescence”, what they want to refer to as "correct composition"? - most appropriate body composition?
References
59 references seems to be a little excessive. It has some older references and less scientific robustness, which can be removed.
Reviewer 3 Report
General comments:
The manuscript topic is of a good practical and scientifical interest and might be a good set for data comparison. The introduction is clear, using almost all relevant literature and creates a good reason to run author’s study. The methodological section allows the study replication and information about the participant is appropriate. Statistical section and result are also clear, especially the canonical analyses seem to be well done. I actually have just some minor suggestion and thus I recommending this article to be published.
Specific comments:
Line 19 - 20: The statement that soccer and basketball are multifactorial team sports is sort of vague. Just state in second sentence that “An aspects that influences performance in soccer and basketball are ….
Line 22-24: there are too many double dots in those sentences. Please re-word this part.
Line 27 – 31: Please put into the text which statistical teste you are referring as results and add some p or effect size values.
Line 32 – 33: the statement „ study confirms the value of sport to achieve an optimal body composition in adolescence“ Is too general and do not relate to the main data (you don’t have measure for optimal, perhaps optimal maturation?). Therefore, skip or revise this sentence.
Line 37: Some keywords are already in the title , use more general one instead.
Line 67: correct the letter font.
Line 138: Which exact Slaughter equation did you use for the body composition and why.
Line 175: Most of the statistics is appropriate, however the significant results should be checked also for effect size.
Discussion line 160: Discuss little more why femur and humerus wide are sport related determinates.
